# Diagnostic Performances of the ACR-TIRADS System in Thyroid Nodules Triage: A Prospective Single Center Study

**DOI:** 10.3390/cancers13092230

**Published:** 2021-05-06

**Authors:** Davide Leni, Davide Seminati, Davide Fior, Francesco Vacirca, Giulia Capitoli, Laura Cazzaniga, Camillo Di Bella, Vincenzo L’Imperio, Stefania Galimberti, Fabio Pagni

**Affiliations:** 1Department of Radiology, ASST Monza, 20900 Monza, Italy; davide.leni@asst-monza.it (D.L.); d.fior@asst-monza.it (D.F.); francesco.vacirca@asst-monza.it (F.V.); 2Department of Pathology, University of Milan—Bicocca (UNIMIB), 20900 Monza, Italy; d.seminati@campus.unimib.it (D.S.); petala.83@tiscali.it (L.C.); camillo.dibella@asst-monza.it (C.D.B.); vincenzo.limperio@gmail.com (V.L.); 3Bicocca Bioinformatics Biostatistics and Bioimaging B4 Center, School of Medicine and Surgery, University of Milan—Bicocca (UNIMIB), 20900 Monza, Italy; giulia.capitoli@unimib.it (G.C.); stefania.galimberti@unimib.it (S.G.)

**Keywords:** ultrasound imaging, thyroid carcinoma, fine-needle aspiration

## Abstract

**Simple Summary:**

On a prospective series of 480 thyroid nodules, the ACR-TIRADS demonstrated a sensitivity and specificity in performing FNA of 58.9% and 59%, respectively. The execution of FNA on nodules with ACR class ≥3 independently from the dimensional criteria would increase the sensitivity to 95% and reduce the false negatives rate (7.3%, 7/96), prompting a re-evaluation of the size criteria. The need for reduction in inappropriate hospital admissions prompts a rigorous triage of patients, and future prospective studies to improve current performances might be considered.

**Abstract:**

Ultrasound scores are used to determine whether thyroid nodules should undergo Fine Needle Aspiration (FNA) or simple clinical follow-up. Different scores have been proposed for this task, with the American College of Radiology (ACR) TIRADS system being one of the most widely used. This study evaluates its ability in triaging thyroid nodules deserving FNA on a large prospective monocentric Italian case series of 493 thyroid nodules from 448 subjects. In ACR 1–2, cytology never prompted a surgical indication. In 59% of cases classified as TIR1c-TIR2, the FNA procedure could be ancillary, according to the ACR-TIRADS score. A subset (37.9%) of cases classified as TIR4-5 would not undergo FNA, according to the dimensional thresholds used by the ACR-TIRADS. Applying the ACR score, a total of 46.5% thyroid nodules should be studied with FNA. The ACR system demonstrated a sensitivity and specificity of 58.9% and 59% in the identification of patients with cytology ≥TIR3A, with a particularly high false negative rate for ACR classes ≥3 (44.8%, 43/96), which would dramatically decrease (7.3%, 7/96) if the dimensional criteria were not taken into account. In ACR 3–4–5, a correspondence with the follow-up occurred in 60.3%, 50.2% and 51.9% of cases. The ACR-TIRADS is a useful risk stratification tool for thyroid nodules, although the current dimensional thresholds could lead to an underestimation of malignant lesions. Their update might be considered in future studies to increase the screening performances of the system.

## 1. Introduction

In first-level general hospitals, ultrasound (US)-guided Fine Needle Aspiration (FNA) is crucial to confirm the benign nature of thyroid nodules. Although this cytological procedure represents an invaluable mini-invasive tool for the assessment of nodule composition, first level hospitals more frequently face a high prevalence of benign, multinodular and hyperplastic goiter cases, and the restriction of FNAs to significant lesions is mandatory in this setting. For this purpose, the development of ancillary tools to evaluate thyroid lesions deserving of cytological assessment could be of help in reducing the rate of unnecessary biopsies. During recent years, different US-based systems, known as Thyroid Imaging Reporting and Data Systems (TIRADS), have been proposed to characterize nodules and obtain a putative indication to FNA [1,2]. Each of these rely on diverse imaging criteria that could be eminently “US-pattern” or point scale guided. Among the “pattern”-based methods, the American Thyroid Association (ATA) and the European Thyroid Association (EU-TIRADS) ones probably represent the most widely employed, and consist in the recognition of high risk and pretty qualitative criteria (such as nodule composition, echogenicity, margins, shape and calcifications), without a quantification scale that can ensure adequate inter-observer reproducibility [3,4,5]. Although these systems are largely employed in different countries, the most widely used US-based system is probably still represented by the American College of Radiology (ACR)-TIRADS one, a point scale option that allows one to readily “quantify” the level of risk for each lesion, representing a reliable alternative with practical feasibility [4,6]. For this reason, and for its relevance in routine clinical practice, this numerical ACR-TIRADS scoring system was selected to evaluate its role as a screening tool in a general hospital with a high prevalence of benign nodules.

## 2. Materials and Methods

### 2.1. Patients

This prospective study included 448 consecutive patients who underwent US-guided FNA from January to June 2019 at the interventional radiology clinic, ASST Monza, Italy, during the final phases of an Italian Association for Research on Cancer (AIRC)-granted project for the diagnosis of thyroid carcinoma. Before being submitted to the US and FNA procedures, all patients were evaluated by an endocrinologist for the assessment of thyroid function and, in cases of hyperthyroidism with associated US-detected nodule, patients were studied with scintigraphy, as indicated by the guidelines [5]. All 493 nodules were subjected to FNA, regardless of their US appearance and/or score, due to the previous specialistic endocrinological clinical indication. For all patients, a period of at least 12 months of follow-up was available. A total of 13 lesions with a TIR1 cytology and no FNA repeating were excluded for a total of 480 nodules, and 435 subjects were considered in the final analysis. The study was approved by the ASST Monza Ethical Board (October 2016, 27102016) and appropriate informed consent was obtained from all patients.

### 2.2. Ultrasound Evaluation

Patients were placed supine with the neck in hyperextension. The US was performed with the Philips Epiq Elite machine. For each nodule, radiologists measured the major axis on real-time clips and analyzed its composition, echogenicity, shape, margins and presence of calcifications, as previously described [6]. The final theoretical indication for FNA execution was formulated according to the ACR algorithm. This system assesses some of the ultrasound features of thyroid nodules, such as composition, echogenicity, shape, margins and echogenic foci, attributing a progressive score to each of these characteristics. The final class assigned to the nodule is based on the sum of these points, with suspicious classes leading to different operative indications based on the dimensional criteria (Table 1).

### 2.3. Cytopathology and Histopathology

Aspiration was performed by two pathologists (FP/CDB), both with ten years of experience in thyroid FNA under US guidance, with 22–25 Gauge diameter needles. The aspirated material was smeared onto 3–4 traditional slides per nodule. The slides were then fixed with spray alcohol (Cytofix, propan-2-ol) and then stained with Papanicolau, or air dried and stained with May-Grunwald Giemsa. Cases were diagnosed according to the 5-tiered Italian SIAPEC (Società Italiana di Anatomia Patologica e Citologia Diagnostica) system for reporting thyroid cytopathology as follows: TIR1 (unsatisfactory; TIR1c cystic; Bethesda I), TIR2 (benign; Bethesda II), TIR3A (indeterminate for malignancy, low risk; Bethesda III), TIR3B (indeterminate for malignancy, high risk; Bethesda IV), TIR4 (suspicious; Bethesda V), TIR5 (malignant; Bethesda VI) [7]. The ACR-TIRADS indication to FNA was considered correct in the presence of a cytology of >TIR3A. TIR1c-TIR2 patients, and those with TIR3 that did not undergo surgery, performed a US examination 12 months after the first US-guided FNA by the same radiologist [8]. Nodules were considered benign in the absence of the following: new echographic malignant features, >20% increase in size, nodes metastasis, and appearance of new suspicious nodules.

Histological evaluation was performed on surgical specimens of total or hemi-thyroidectomy, carried out according to the Italian consensus guidelines [9]. The tissue was formalin fixed, paraffin embedded and stained with haematoxylin and eosin.

### 2.4. Statistical Analysis

Mean and standard deviation or quartiles were used for descriptive purposes, as appropriate. The diagnostic ability of ACR-TIRADS in distinguishing thyroid nodules that did or did not require FNA was evaluated using cytology as a reference standard. Results of the ACR-TIRADS classification were also compared to the histopathological findings for nodules undergoing surgery or follow-up information at 12 months for non-resected nodules. Sensitivity, specificity, positive and negative predictive values (PPV and NPV) were determined alongside their 95% confidence intervals (CI). The agreement between the dimensions of the same nodule provided by two radiologists was assessed with the Bland-Altman approach. Finally, an UpSet plot was used to show the occurrence of all possible combinations of ACR-TIRADS components observed in the thyroid nodules. All the statistical analyses were performed using the open-source R software v.3.6.0.

## 3. Results

From the initial cohort of 493 nodules from 448 consecutive patients who underwent US-guided FNA, 13 were excluded from the comparative analysis due to an inadequate cytological class (TIR1) without FNA repetition. The final cohort consisted of 480 nodules from 435 patients. Patients’ age ranged between 18 and 90 years, with a mean of 59 years (sd = 14.93), and 73% (327) were female. The prebiopsy endocrinological assessment of thyroid function demonstrated that the vast majority of the patients were in a state of euthyroidism (92%), with a minority presenting with overt or subclinical hypothyroidism (7%) or hyperthyroidism (1%). This last group also underwent radionuclide thyroid scintigraphy; none of them demonstrated a hyperfunctioning (“hot”) state of the nodules. A background of multinodular goiter was recorded in 121 patients (27%). Nodule size ranged from 5 to 70 mm (median 18 mm, I–III quartiles 12–27 mm). A subset of patients underwent surgery (*n* = 49), 34 of whom (7%) had a final diagnosis of malignancy on histology (Appendix A). The most frequent diagnosis in this group was papillary thyroid carcinoma (27/34, 79%).

### 3.1. ACR-TIRADS System Performances

SIAPEC/Bethesda diagnostic classes and the corresponding ACR-TIRADS score are reported in Table 2.

None of the cases classified as ACR-TIRADS 1–2 underwent surgery, as per cytological indication (0/69, Figure 1).

In the other ACR classes, the attributed ACR score that would indicate the performance of FNA was mainly affected by nodule diameter. Of the cases with an ACR class ≥3 not reaching the dimensional criteria for biopsy, and therefore without a strict FNA indication as per ACR but with a final cytological class of TIR3A/III-AUS (*n* = 36) or TIR3B/IV-SFN (*n* = 6), a subset underwent surgery (14%, 6/42, of which 2 TIR3A/III-AUS and 4 TIR3B/IV-SFN), with a final histological diagnosis of malignancy in one case, classified as TIR3A/III-AUS, and corresponding to an encapsulated follicular variant of papillary thyroid carcinoma (EFVPTC).

Concordance among the radiological and cytological results (FNA result ≥ TIR3A) was found in 82 out of 136 (60.3%) ACR-TIRADS 3 nodules, as well as in 111 out of 221 (50.2%) ACR-TIRADS 4 and 28 out of 54 (51.9%) ACR-TIRADS 5 nodules (Figure 2).

In 59% (197 out of 334) of cases classified as TIR1c-TIR2, the FNA procedure could be ancillary according to the ACR-TIRADS score. On the other hand, a subset of cases (11 out of 29, 37.9%) classified as TIR4-5 would not undergo FNA according to the dimensional thresholds used by the ACR-TIRADS. This dimensional cut-off would also have excluded 43 TIR3A (44.8%) and 6 TIR3B (28.6%), for a total of 60 out of 146 (41%) excluded nodules with an FNA result of ≥ TIR3A.

In summary, when applying the ACR-TIRADS score, in 46.5% of the nodules of this series (223/480), there was a FNA indication, 86 of which revealed clinically significant cytological results (≥TIR3A). The ACR score provided appropriate indications for the FNA execution, with a 58.9% (86/146, 95% CI = 50.5–67.0) sensitivity, a 59% (197/334, 95% CI = 53.5–64.3) specificity, 76.7% (197/257, 95% CI = 71.0–81.7) NPV and 38.6% (86/223, 95% CI = 32.1–45.3) PPV (Figure 3).

The same evaluation performed while excluding nodule diameter from the calculation led to a marked increase in sensitivity to 95.2% (139/146, 95% CI = 90.4–98.1) and NPV to 89.9% (62/69, 95% CI = 80.2–95.8), as well as a relative drop in specificity to 18.6% (62/334, 95% CI = 14.5–23.2). ACR sensitivity ranged from 66% (33/50) for FNA ≥ TIR3B to 70.6% (12/17) for TIR5 alone.

The comparison of inter-operator variability in the assessment of 17 nodules classified as TIR1, and repeated by two different radiologists, revealed only one relevant disagreement (ACR class 5 vs. 1, Appendix A). Moreover, some minor discrepancies were found in the assessment of shape (*n* = 1), composition (*n* = 3), echogenic foci (*n* = 2) and echogenicity (*n* = 8). A Bland–Altman analysis of interobserver agreement for the assessment of the nodule size demonstrated a good global agreement, with a bias of 0.53 mm and limits of agreement equal to −7.8 mm and 8.8 mm (Appendix A).

During the available follow-up, none of the cases classified as TIR1c and TIR2 acquired any of the suspicious US features reported in Methods, thus suggesting the benign nature of the lesion, as per previous recommendations [8]. On the other hand, histological evaluation was carried out on 10.2% of nodules (49/480), the majority of which (81.6%) belonged to the group of cases with an FNA result of ≥TIR3B (40/49). The patients with a TIR3A class who did not undergo surgery were followed-up with the US, and a minority of these (17/96) showed modification of either nodule shape or dimension, leading to a repeat biopsy, with confirmation of TIR3A class (*n* = 10) or downgrading to the TIR2 class (*n* = 7). Considering the final histology or the follow-up, the ACR-TIRADS and the SIAPEC system demonstrated a sensitivity of 67.6% (23/34, 95% CI = 49.5–82.6) and 94.1% (32/34, 95% CI = 80.3–99.3), and a specificity of 57.2% (210/367, 95% CI = 52.0–62.3) and 97.5% (358/367, 95% CI = 95.4–98.9), respectively (Table 3).

### 3.2. Ecographic Characteristics

The detailed US features that concurred to the formulation of ACR-TIRADS score, and their correlation with the final SIAPEC/Bethesda class, are reported in Table 4.

The analysis per single radiologic feature demonstrated a prevalence of solid composition in those cases (81.8%, 112/137) classified as benign by FNA (<TIR3A), but without a relevant difference among nodules with negative and suspicious/positive cytology (81% vs. 94%, respectively). Cystic composition, even if rare (4%, 19/480), was a reliable benign characteristic, being associated only with the TIR2 class, as was noted with the spongiform appearance (83% diagnosed as TIR2). For echogenicity, nodules were perceived mainly as hypoechoic (268, 55.9%) and isoechoic (158, 32.9%); specifically, hypoechoic nodules did not demonstrate a significant impact on FNA indication (55.2% sensitivity, 148/268), as well as on the cytological outcome (32% sensitivity, 86/268). The wider-than-tall shape was predominant, and cases with this feature frequently underwent FNA (92.9%, 26/28) for its high ACR-TIRADS score (i.e., 3). Of the 41 nodules with irregular margins, 32 (78.0%) had a diameter greater than the ACR cut-off for the FNA execution, although only 21 (51.2%) showed relevant final results (≥TIR3A) after cytology. Thyroid lesions mainly presented none or large comet-tail artifacts, with few microcalcifications or peripheral calcifications. Microcalcifications, which are associated with three points in the ACR system, were found in 28 lesions of our series, and 16 (57.1%) were significant after cytology (≥TIR3A).

Overall joint distribution of the five ACR-TIRADS components is described in Figure 4.

Two specific US patterns, obtained by the combination of different nodule features, were more frequently represented in this cohort. The first one (characterized by solid, hypoechoic, wider-than-tall, smooth margins appearance, without calcifications) was noted in 133 (27.7%) lesions, with another 90 cases (18.8%) showing similar characteristics, with differences in echogenicity (isoechoic/hyperechoic instead of hypoechoic) representing the second leading pattern. Dividing the cohort into four subgroups based on the final SIAPEC/Bethesda class and the FNA indication, as per ACR-TIRADS score (Table 3), the two described US patterns confirmed their higher frequency in the cohort. Considering the final histology or follow-up, the presence of microcalcifications or taller-than-wide shape was strongly related with malignancy (43.5% in malignant vs. 25.2% in benign), whilst the combination of solid, isoechoic/hyperechoic, wider-than-tall, and smooth margins without calcifications was prevalent in benign cases (37.2% in benign vs. 18.2% malignant).

## 4. Discussion

The performance of US-guided FNA on thyroid nodules represents an invaluable tool for selecting lesions that would benefit from surgery [6,10]. As opposed to the high prevalence of carcinomas generally found in second level centers receiving high-risk patients, the most frequent scenario encountered in first level hospitals is often characterized by a multinodular goiter background, in which an appropriate FNA screening is mandatory. Although the bioptic procedure is safe with rare complications, the application of TIRADS scores as an ancillary tool may improve the triage with a more efficient clinical management, postponing the performance of FNA in low-risk cases [11,12]. The ACR-TIRADS method was chosen in our study due to its putative superior overall performance [4,13,14,15,16]. Indeed, the other available and widely used US-based evaluation systems (ATA and EU-TIRADS) mainly rely on the assessment of nodules’ features (e.g., shape, margins and echogenicity) without the attribution of a score for each of these characteristics, representing purely pattern-based classifications. On the other hand, in the ACR-TIRADS, the different components included in the algorithm are scored with specific weights, promptly quantifying with numbers the US features evaluated by the radiologist [6]. This type of approach is particularly useful in thyroid glands containing multiple nodules, in which the cytological characterization of the dimensionally prevalent one failed to demonstrate a significant impact on a patient’s outcome [17]. Moreover, although all the three systems take in to account a “dimensional” threshold for the ≥3 classes (suspicious nodules), determining the final operative indication (whether to perform FNA or not), in the ACR classification, the size criteria are more detailed and stringent, further stratifying the lesions that could benefit from a cytological biopsy (Appendix A).

In the present case series, a prevalence of female patients 73% (327) was noted, as already reported in the literature (61.2–94.9%), as well as a high frequency of goiters 27% (121), confirming the data obtained by a previous study performed in northern Italy (36%), with a possible indirect impact on the relatively low incidence (30.4%) of ≥TIR3A lesions [18,19]. The prevalence of malignancy was 7%, with ranges in the literature between 3.9% and 66.1% [18]. The sensitivity and specificity of the cytology assessment were 94.1% and 97.5%, respectively, which is coherent with the literature data (50–94% and 32–100%) [20]. A good correspondence between radiology and cytology was noted in cases with ACR class 1–2, 90% (62/69) of which belonged to the <TIR3A (Bethesda AUS) group, and only seven nodules were classified as TIR3A, benefitting from a strict follow-up policy (Table 2). These results support an excellent “rule-out” role for the ACR-TIRADS system that would significantly reduce the number of FNAs performed on benign nodules, as previously demonstrated [13,14,15,16,18,21,22,23,24]. For these cases, FNAs may be avoided if not strictly indicated by clinical needs, such as symptomatic goiter or therapeutic decompression of upper airways. On the other hand, in 44.8% (43/96) of nodules classified as TIR3A, the execution of FNA was not indicated following the ACR-TIRADS score and, in many of these cases (83.7%, 36/43), the nodule dimension would be the main limiting factor in biopsying the lesion. However, this “false negative” rate could be overestimated, considering the relatively intrinsic low risk-of-malignancy of most TIR3A lesions [15,25,26]. On this point, it should be noted that, although representing the equivalent of an intermediate class for the Bethesda system, and thus introduced as an effort in levelling out the different reporting systems to avoid grey zones or doubtful indications, the criteria to assign a TIR3A or TIR3B SIAPEC class do not perfectly match the corresponding III/AUS-FLUS and IV/SFN class of the Bethesda system [26]. As opposed to the AUS definition, which is mainly based on architectural and/or nuclear atypia insufficient to be suspicious for malignancy, the extensors of the SIAPEC system intended to base the class definitions on the concept of “risk of malignancy” as the pivotal parameter of the proposal. As a consequence, the TIR3A class does not completely overlap with the AUS or FLUS categories, being represented by follicular proliferations with a mild degree of atypia or low malignant risk, by definition. On the other hand, a more reliable correspondence can be noted between the TIR3B and SFN Bethesda category, with both showing a higher prevalence of pure follicular proliferations or Hurthle-cells type neoplasms, actually leading to a surgical indication for the distinction between adenomas and carcinomas. Thus, the discrimination of AUS and FLUS within the SIAPEC TIR3A and TIR3B categories cannot be reliably performed, as debated in the literature [25,26].

Extending the FNA indications to all nodules with ACR-TIRADS ≥3, independent of the dimensional criteria, false negatives would be reduced to 7.3% (7/96), with a relative increase in sensitivity and NPV from 59% to 95% and from 76% to 90%, respectively. The elimination of the dimensional thresholds in TIR3B (SFN Bethesda), TIR4 (suspicious Bethesda class V) and TIR5 (malignant) classes would eliminate false negatives, with a consequent drop in specificity from 58% to 18%. A reasonable approach to this problem would be the activation of US surveillance for nodules with ACR-TIRADS ≥3, and a size below the threshold for FNA indication, postponing the eventual cytological evaluation to subsequent checks [6]. Middleton et al. recently found that the modification of the size cutoff to decrease the number of missed malignancies would mainly lead to an increase in the number of biopsied benign nodules, without a significant impact on the prognosis of malignant ones [27]. Conversely, another study underlined that, globally, the ACR classification system may be particularly restrictive, with a possible delay in diagnosis or therapy for carcinomas [28].

The analysis of single radiologic features that contribute to the final score demonstrated that nodules with microcalcifications, or taller-than-wide shape, are more frequently malignant in nature, which is in agreement with the high-suspicious rate previously reported [6]. Our findings also suggest the possibility of a future revision of the weights attributed to some ACR-TIRADS system features, with an eventual reduction in the scores associated with solid composition and hypoechogenicity (2 points each), whose impact on the screening process was not particularly relevant in this study (Table 3). Moreover, future improvements in the algorithm might include alternative characteristics, such as elastography, that already demonstrated a potential impact in classification systems used for other organs and districts [29,30,31,32]. As per previous studies relying on the ACR-TIRADS class, FNA would be indicated in a range of 17% to 40%; in this paper, cases with FNA indicated by the ACR system accounted for 46.5% of the nodules, confirming its high “rule-out” role. The ancillary FNA rate was 28.5%, which is significantly lower than the results reported in the literature (63.8%) [4]. Some limitations of our study should be noted. First, all the nodules were subjected to FNA following a previous specialistic endocrinological clinical indication, independently from the ACR-TIRADS criteria. Moreover, the available minimum follow-up of this study (12 months) was not adequate to assess the benign behavior of a thyroid nodule, since some thyroid cancers are characterized by a potentially very slow growth. Another limiting factor is represented by the restricted number of cases that have post-surgical histological assessment available, partly affecting the evaluation of predicting performances of the systems that, hence, need to be confirmed by a larger series. A comparison of the performances of different US classification systems, which could be point scale (ACR) or pattern-based (EU), will be more comprehensively addressed in the future with a dedicated effort. Finally, a comparison of performances between radiologists was evaluated in a small subgroup of 17 nodules.

## 5. Conclusions

In this study, the ACR-TIRADS score was able to identify those nodules that can be excluded from the FNA based on an easy-to-recognize combination of US benign aspects. On the other hand, a high false negative rate was observed for ACR classes ≥3, mainly affected by the presence of a dimensional threshold that leads it to exclude the execution of FNA in otherwise deserving nodules. An update of the size criteria, and of the US features points, might be considered in future studies to increase screening performances of the ACR-TIRADS system.

## Figures and Tables

**Figure 1 cancers-13-02230-f001:**
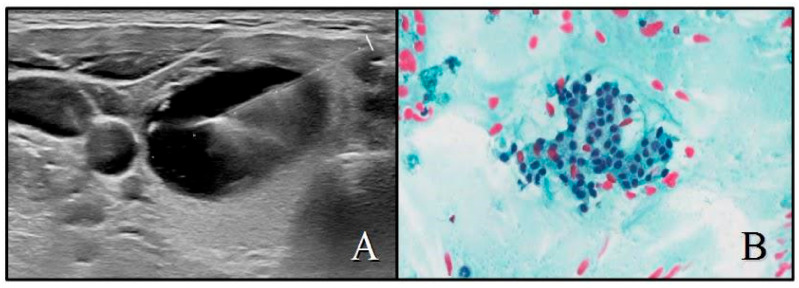
(**A**) Example of ACR-TIRADS class 1 nodule: cystic (0 points), anechoic (0 pts), wider-than-tall (0 pts), smooth margins (0 pts), no echogenic foci (0 pts), with a total ACR score of 0 points. Diameter 35 mm; (**B**) SIAPEC system cytological class TIR2 (PAP, ×20).

**Figure 2 cancers-13-02230-f002:**
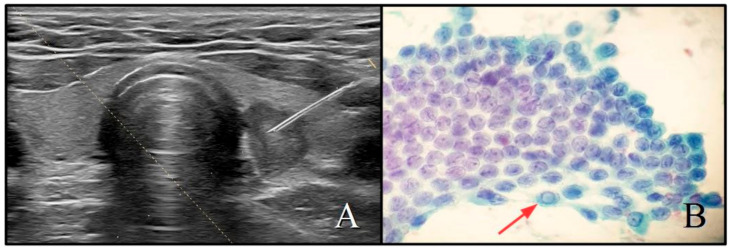
(**A**) Example of ACR-TIRADS class 5 nodule: solid (2 points), very hypoechoic (3 pts), taller-than-wide (3 pts), irregular margins (2 pts), no echogenic foci (0 pts), with a total ACR score of 10 points. Diameter 14 mm; (**B**) SIAPEC system cytological class TIR5: there are several grooves and an evident nuclear pseudoinclusion (red arrow) (PAP, ×40).

**Figure 3 cancers-13-02230-f003:**
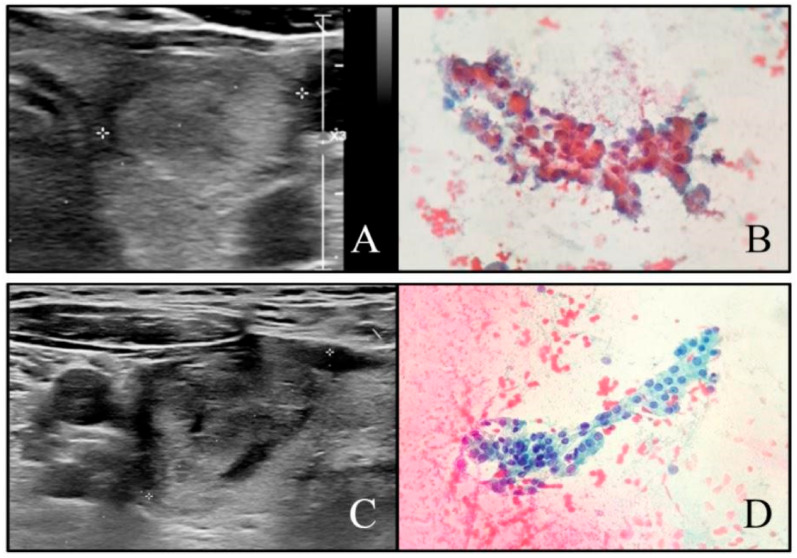
(**A**) Example of false negative case. ACR-TIRADS class 3 nodule: solid (2 points), isoechoic/hyperechoic (1 pt), wider-than-tall (0 pts), ill-defined margins (0 pts), no echogenic foci (0 pts), with a total ACR score of 3 points. Diameter 14 mm. (**B**) SIAPEC system cytological class TIR3B: there is a predominant Hurthle metaplasia; therefore, the nodule has received a surgical indication (PAP, ×20). At the histological evaluation, the nodule turned out to be a Hurthle cell adenoma. (**C**) Example of false positive case. ACR-TIRADS class 4 nodule: solid (2 points), isoechoic/hyperechoic (1 pt), taller-than-wide (3 pts), ill-defined margins (0 pts), no echogenic foci (0 pts), with a total ACR score of 6 points. Diameter 2.5 mm. (**D**) SIAPEC system cytological class TIR2 (PAP, ×20).

**Figure 4 cancers-13-02230-f004:**
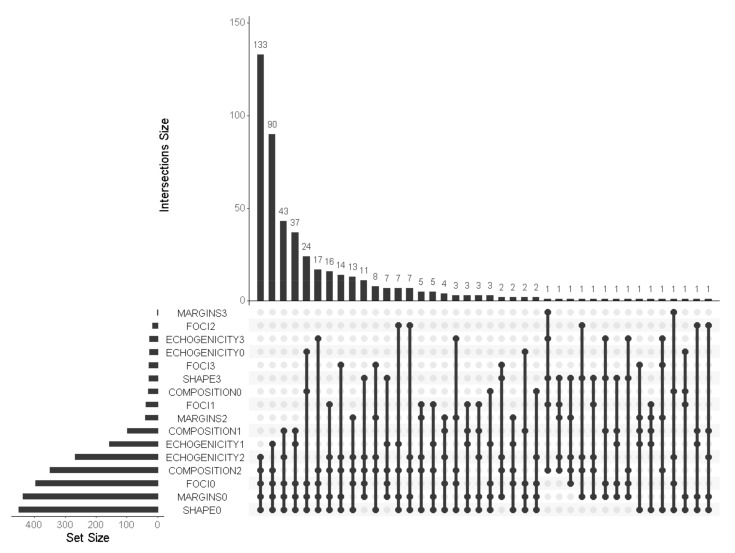
UpSet plot showing the ecograpich ACR-TIRADS components of the 480 thyroid nodules: every possible intersection is represented by the bottom plot; their occurrence is shown on the top barplot (intersection size); the total size of each set is represented on the left barplot (set size).

**Table 1 cancers-13-02230-t001:** Ultrasound criteria to assess thyroid nodules according to the ACR-TIRADS system.

**Ecographic Characteristics**		**Points**
Composition	Cystic or almost completely cystic	0
	Spongiform	0
	Mixed cystic and solid	1
	Solid or almost completely solid	2
Echogenicity	Anechoic	0
	Hyperechoic or isoechoic	1
	Hypoechoic	2
	Very hypoechoic	3
Shape	Wider-than-tall	0
	Taller-than-wide	3
Margins	Smooth	0
	Ill-defined	0
	Lobulated or irregular	2
	Extrathyroidal extension	3
Echogenic foci	None or large comet-tail artifacts	0
	Macrocalcifications	1
	Peripheral (rim) calcifications	2
	Punctate echogenic foci	3
ACR Total Score	ACR Classes	Final Indication
0	ACR1: Benign	No FNA
2	ACR2: Not suspicious	No FNA
3	ACR3: Mildly suspicious	FNA if ≥2.5 cm
		Follow if ≥1.5 cm
4 to 6	ACR4: Moderately suspicious	FNA if ≥1.5 cm
		Follow if ≥1 cm
>7	ACR5: Highly suspicious	FNA if >1 cm
		Follow if ≥0.5 cm

The presence of multinodular goiter, i.e., an enlarged gland with at least 2 nodules, was annotated too. US was performed by 3 different radiologists (DL, DF, FV), experts in thyroid imaging, who equally contributed to the evaluation of the case series. ACR: American College Radiology; FNA: Fine Needle Aspiration

**Table 2 cancers-13-02230-t002:** ACR-TIRADS with SIAPEC/Bethesda system cytological classes. False negative cases are in bold, false positive cases are in italics.

SIAPEC System							
ACR-TIRADS	TIR1c	TIR2 (II)	TIR3A (III/aus)	TIR3B (IV/sfn)	TIR4 (V/susp)	TIR5 (VI)	Total
No indication to FNA							
ACR 1	12	16	2	0	0	0	30
ACR 2	1	33	5	0	0	0	39
ACR 3 < 2.5 cm	0	63	13	2	2	1	81
ACR 4 < 1.5 cm	0	72	21	4	3	4	104
ACR 5 < 1.0 cm	0	0	2	0	1	0	3
Indication to FNA							
ACR 3 ≥ 2.5 cm	1	35	15	3	1	0	55
ACR 4 ≥ 1.5 cm	1	77	29	7	3	0	117
ACR 5 ≥ 1.0 cm	0	23	9	5	2	12	51
Total	15	319	96	21	12	17	480

**Table 3 cancers-13-02230-t003:** Histology or follow-up classification and (**A**) ACR-TIRADS; (**B**) SIAPEC/Bethesda system. Data are reported as *n* and (%). In all, 70 TIR3A, 8 TIR3B and 1 TIR4 nodules were lost to follow-up.

**(A) ACR-TIRADS**	**Histology or Follow-Up**		**Total**
	**Benign**	**Malignant**	
ACR 1	29 (7.9)	0 (0.0)	29 (7.2)
ACR 2	35 (9.5)	0 (0.0)	35 (8.7)
ACR 3 < 2.5 cm	67 (18.3)	4 (11.8)	71 (17.7)
ACR 4 < 1.5 cm	79 (21.5)	7 (20.6)	86 (21.5)
ACR 5 < 1.0 cm	0 (0.0)	0 (0.0)	0 (0.0)
ACR 3 ≥ 2.5 cm	44 (12.0)	1 (2.9)	45 (11.2)
ACR 4 ≥ 1.5 cm	86 (23.4)	4 (11.8)	90 (22.5)
ACR 5 ≥ 1.0 cm	27 (7.4)	18 (52.9)	45 (11.2)
Total	367 (100.0)	34 (100.0)	401 (100.0)
(B) SIAPEC/ Bethesda Systems	Histology or Follow-Up		Total
	Benign	Malignant	
TIR1c	15 (4.1)	0 (0.0)	15 (3.7)
TIR2 (benign)	319 (86.9)	0 (0.0)	319 (79.6)
TIR3A (aus)	24 (6.5)	2 (5.9)	26 (6.5)
TIR3B (sfn)	8 (2.2)	5 (14.7)	13 (3.3)
TIR4 (suspicious)	1 (0.3)	10 (29.4)	11 (2.7)
TIR5 (malignant)	0 (0.0)	17 (50.0)	17 (4.2)
Total	367 (100.0)	34 (100.0)	401 (100.0)

**Table 4 cancers-13-02230-t004:** Distribution of the ecographic characteristics in the ACR-TIRADS system by SIAPEC/Bethesda and ACR-TIRADS classes. The ACR-TIRADS scores corresponding to each modality are reported in parenthesis. Data are reported as *n* and (%).

Ecographic Characteristic	TIR < 3A	(<AUS)	TIR ≥ 3A	(≥AUS)	Total
	ACR-FNA No	ACR-FNA Yes	ACR-FNA No	ACR-FNA Yes	
Composition					
Cystic (0)	19 (9.6)	0 (0.0)	0 (0.0)	0 (0.0)	19 (4.0)
Spongiform (0)	10 (5.1)	0 (0.0)	2 (3.3)	0 (0.0)	12 (2.5)
Mixed (1)	60 (30.5)	25 (18.2)	9 (15.0)	5 (5.8)	99 (20.6)
Solid (2)	108 (54.8)	112 (81.8)	49 (81.7)	81 (94.2)	350 (72.9)
Echogenicity					
Anechoic (0)	25 (12.7)	0 (0.0)	2 (3.3)	0 (0.0)	27 (5.6)
Isoechoic/hyperechoic (1)	78 (39.6)	38 (27.7)	21 (35.0)	21 (24.4)	158 (32.9)
Hypoechoic (2)	89 (45.2)	93 (67.9)	31 (51.7)	55 (64.0)	268 (55.9)
Marked hypoechoic (3)	5 (2.5)	6 (4.4)	6 (10.0)	10 (11.6)	27 (5.6)
Shape					
Wider-than-Tt all (0)	195 (99.0)	122 (89.1)	60 (100.0)	75 (87.2)	452 (94.2)
Taller-than-wide (3)	2 (1.0)	15 (10.9)	0 (0.0)	11 (12.8)	28 (5.8)
Margins					
Smooth (0)	132 (67.0)	65 (47.4)	31 (51.7)	49 (57.0)	277 (57.7)
Ill-defined (0)	60 (30.5)	57 (41.6)	25 (41.7)	19 (22.1)	161 (33.6)
Irregular (2)	5 (2.5)	15 (10.9)	4 (6.7)	17 (19.8)	41 (8.5)
Extra-thyroidal (3)	0 (0.0)	0 (0.0)	0 (0.0)	1 (1.2)	1 (0.2)
Echogenic Foci					
None or large comet-tail artifacts (0)	182 (92.4)	103 (75.2)	54 (90.0)	57 (66.3)	396 (82.5)
Macrocalcifications (1)	11 (5.6)	13 (9.5)	3 (5.0)	11 (12.8)	38 (7.9)
Peripheral calcifications (2)	4 (2.0)	9 (6.6)	0 (0.0)	5 (5.8)	18 (3.8)
Punctate echogenic foci (3)	0 (0.0)	12 (8.8)	3 (5.0)	13 (15.1)	28 (5.8)
Total	197 (100.0)	137 (100.0)	60 (100.0)	86 (100.0)	480 (100.0)

## Data Availability

The authors declare that they had full access to all of the data in this study, and the authors take complete responsibility for the integrity of the data and the accuracy of the data analysis.

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
