# Peer review of "Diagnostic Performances of the ACR-TIRADS System in Thyroid Nodules Triage: A Prospective Single Center Study"

_cancers, 2021, doi:10.3390/cancers13092230_

Round 1

Reviewer 1 Report

The Authors present a prospective single center study with the aims to evaluate the role of the ACR-TIRADS a screening tool in the diagnostic work-up of thyroid nodules. In my view, the manuscript is of interest to the clinical community; however, the presentation of the study results with respect to ACR-TIRADS methodology is not presented in a clear and comprehensive, clinical meaningfull manner. To my understanding, the most important finding of the study is that patients with ACR classification 1 and 2 rarely exhibit cytology scores ≥3, i.e. highlighting the role of the suggested ultrasonographic classification system to rule-out unnecessary FNAs. The Authors need to present the advantages of the proposed methodology over TIRADS and EU-TIRADS. Otherwise, this seems to be another study correlating thyroid US classification to cytology.

Major points:

  1. Please expand on the study hypothesis and the introduction section.
  2. Please use standard nomenclature folowing the STROBE guideline to report the study results.
  3. Please elaborate on the statistical methodology used in this study in detail.
  4. Please briefly define ACR-TIRADS in the Methods section. This could probably explain why the Authors apply the 1cm, 1.5cm and 2.5cm cut-offs in Table 1. Please discuss how this is different from TIRADS and EU-TIRADS in the Discussion section.
  5. Page 3. Results section. Please provide mean +/-SD or median with range as appropriate.
  6. One of the most important flaws in the design of this study is that patients included in this series with an available surgical specimen for histological evaluation were only 49 cases. In my view, histopathology should be the golden reference standard. Hence, clinical inferences of the study findings have to be considered with caution, as there is a wide range of discrepancies between the cytological classification (Bethesda, SIAPEC etc) and the final histopathological diagnosis at different institutions.
  7. 44 out of 117 patients with TIR3a and TIRb in Table 1 had no indication for FNA. How many of these were operated based on the findings of FNA and what was the final histopathology in these cases? Please compute sensitivity and specificity in this setting.
  8. With respect to SIAPEC please specify within TIR3A and TYRIB, how many cases had Atypia of Undetermined Significance (AUS) or Follicular Lesion of Undetermined Significance (FLUS) and how correlated this to ACR-TIRADS findings.
  9. The sensitivity and specificity of ACR-TIRADS to identify patients with cytology ≥TIR3a is rather low in this series with a particularly high false negative rate being observed for ACR classes ≥3 . The exact figures should be stated in the abstract and correlation of ACT-TIRADS with final histopathology should also be presented in the subset of patients subjected to surgery.
  10. The follow-up (≥ 12 months) classification of benign vs. malignant is rather short and arbitrary and may be subjected to bias. Please consider presenting data on patients with available histopathology seperately.
  11. In Figure 4 the headings are not clear and the Reviewer cannot assess neither the x axis nor the y axis.
  12. The Authors state in the Discussion section that ”The sensitivity and specificity of the SIAPEC system were 94.1% and 97.5% respectively, coherent with literature data (93-100% and 54-71%)”. Is this computed from the dataset of the study and applies to the institutional sensitivity and specificity of the SIAPEC? It should then be moved then to the Results section.
  13. Please expand on the study limitations in the discussion section.

Author Response

The Authors present a prospective single center study with the aims to evaluate the role of the ACR-TIRADS a screening tool in the diagnostic work-up of thyroid nodules. In my view, the manuscript is of interest to the clinical community; however, the presentation of the study results with respect to ACR-TIRADS methodology is not presented in a clear and comprehensive, clinical meaningfull manner. To my understanding, the most important finding of the study is that patients with ACR classification 1 and 2 rarely exhibit cytology scores ≥3, i.e. highlighting the role of the suggested ultrasonographic classification system to rule-out unnecessary FNAs. The Authors need to present the advantages of the proposed methodology over TIRADS and EU-TIRADS. Otherwise, this seems to be another study correlating thyroid US classification to cytology.

We are glad you appreciated the clinical relevance of our work. Actually you readily struck the point that mainly emerged from our experience, which is represented by the potential usefulness of this system to “rule-out” benign nodules to avoid unnecessary FNAs, especially in the setting of a high prevalence of benign nodules. We agree with you that more space can be dedicated to the comparison with the other (namely ATA and EU) TIRADS systems, and for this reason and to address the other comments which are exposed later on, we expanded the DISCUSSION section on the point. 

Major points:

  1. Please expand on the study hypothesis and the introduction section.

Thanks for your suggestion. We appropriately expanded the introduction section clearly explaining the relevance and the hypothesis that represents the rationale of the proposed study

2. Please use standard nomenclature following the STROBE guideline to report the study results.

Thanks for your suggestion. We followed the STROBE guideline to report the results of the study in this revised manuscript, hoping that it improves the readability and makes it more straightforward.

3.Please elaborate on the statistical methodology used in this study in detail.

According to the reviewer suggestion, we provided further details on the statistical methods used in this study.

4. Please briefly define ACR-TIRADS in the Methods section. This could probably explain why the Authors apply the 1cm, 1.5cm and 2.5cm cut-offs in Table 1. Please discuss how this is different from TIRADS and EU-TIRADS in the Discussion section.

Thanks for your adequate observation. We addressed your comment by adding in the Methods section a description of the ACR score system referring to a new table (TABLE 1) that we added to facilitate the visual understanding of this US-scale and stressing the impact of different dimensional cut-offs that determine different operative indications. Moreover, in Discussion we briefly reported the differences among ACR, ATA and EU-TIRADS systems by adding a comparative table in Supplementary material (Supplementary table 3) that synthesizes what is reported in discussion.

5. Page 3. Results section. Please provide mean +/-SD or median with range as appropriate.

Thanks for your observation we appropriately addressed your comment by adding SD and IQR, or range, where adequate

6. One of the most important flaws in the design of this study is that patients included in this series with an available surgical specimen for histological evaluation were only 49 cases. In my view, histopathology should be the golden reference standard. Hence, clinical inferences of the study findings have to be considered with caution, as there is a wide range of discrepancies between the cytological classification (Bethesda, SIAPEC etc) and the final histopathological diagnosis at different institutions.

Thanks for your observation. We completely agree with your point of view, histology should be the gold standard reference to assess the diagnostic performances of a test to predict the benign or malignant nature of a lesion. However, as we declared in the end part of Introduction, we tried to investigate the possible impact of ACR-TIRADS implementation in the routine assessment of nodules in a center with high prevalence of BENIGN thyroid lesions, which hence do not require generally a surgical approach unless there are compression problems due to the enlargement of the nodules. To address your comment we specified that some of the results could be limited by the restricted number of cases with available histological confirmation adding this as a limitation of the study at the end of the DISCUSSION.Moreover, from the methodological point of view, the strategy to mix histology and follow-up information to compose the gold-standard is always considered in situation, like the one of our study, in which (luckily) surgery is not the unique option.

7. 44 out of 117 patients with TIR3a and TIRb in Table 1 had no indication for FNA. How many of these were operated based on the findings of FNA and what was the final histopathology in these cases? Please compute sensitivity and specificity in this setting.

Thanks for this precious input. Actually, of the 49 patients with TIR3A or TIR3B and no indication for FNA as per ACR criteria, none with ACR 1-2 (7 cases with TIR3A cytology) underwent surgery. Of those with an ACR>3 but without indication as per dimensional criteria, which were 36 TIR3A and 6 TIR3B, a subset underwent surgery (14%, 6/42, 2 TIR3A and 4 TIR3B) with a final diagnosis of malignancy in 1 case (TIR3A, EFVPTC).

Due to the limited number of cases in these groups, the calculation of the sensitivity and specificity is not statistically robust and thus be misleading.

8. With respect to SIAPEC please specify within TIR3A and TYRIB, how many cases had Atypia of Undetermined Significance (AUS) or Follicular Lesion of Undetermined Significance (FLUS) and how correlated this to ACR-TIRADS findings.

As suggested by the reviewer, a general good opportunity is related with the effort in levelling out the different reporting systems to avoid grey zones or doubtful indications. However, as just largely debated in previous papers ( Bongiovanni et al. Cancer Cytopathol 2012 Apr 25;120(2):117-25; Prada et al. Cytopathology. 2014 Jun;25(3):170-6. doi: 10.1111/cyt.12085. Epub 2013 Aug 1), the criteria to assign a TIR3A or TIR3B SIAPEC class don’t perfectly match the corresponding  III/AUS-FLUS and IV/SFN class of the Bethesda system. Actually, in the Bethesda system, the AUS definition includes cell types with architectural and/or nuclear atypia that are more pronounced than those observed in benign lesions, yet not sufficient to be classified as suspicious for malignancy. A similar definition has not been employed in the Italian system deliberately, since the extensors of the SIAPEC system intended to introduce the concept of “risk of malignancy” as the pivotal parameter of the proposal. So the corresponding SIAPEC TIR3A class doesn’t completely overlap the AUS or FLUS categories, being represented by follicular proliferations wild mild degree of atypia or low malignant risk by definition. For the TIR3B cases a stronger correspondence with the SFN Bethesda category can be matched due to the prevalence of pure follicular proliferations or Hurthle-cells type neoplasms, which actually leads to a surgical indication for the distinction between adenomas and carcinomas. Thus, the distinction of AUS and FLUS within th SIAPEC TIR3A and TIR3B categories cannot be reliably performed. We included a paragraph with the appropriate references to the exposed literature debate on the subject, hoping to enrich the interest of the readers about this intriguing topic.

9. The sensitivity and specificity of ACR-TIRADS to identify patients with cytology ≥TIR3a is rather low in this series with a particularly high false negative rate being observed for ACR classes ≥3 . The exact figures should be stated in the abstract and correlation of ACT-TIRADS with final histopathology should also be presented in the subset of patients subjected to surgery.

We addressed your comment expliciting the values you mentioned in the ABSTRACT:

“The ACR-TIRADS system demonstrated a sensitivity and specificity of 58.9% and 59% in the identification of patients with cytology ≥TIR3A and a particularly high false negative rate for ACR classes ≥3 (44.8%, 43/96) that would dramatically decrease (7.3%, 7/96) if the dimensional criteria are not taken into account.” 

Moreover, for your second observation, the small number of patients that underwent surgery (49 cases) does not allow a reliable statistical analysis of the correlation between the ACR class and the final histological diagnosis. For this reason, we extended this type of analysis to the sum of patients with either available surgical specimen or an adequate follow-up time of 12 months without suspect features of the nodule on US or FNA at re-evaluation, considering in this latter instance the nodule as BENIGN, following previous recommendation (“In asymptomatic nodules with a repeated (12-months) benign cytology and no suspicious clinical or US features, routine follow-up may be avoided”, BEL 3, GRADE B; https://doi.org/10.4158/ep161208.gl).

10. The follow-up (≥ 12 months) classification of benign vs. malignant is rather short and arbitrary and may be subjected to bias. Please consider presenting data on patients with available histopathology seperately.

As stated in the answer to the previous comment, the chosen minimum follow-up (12-months) has been already proposed as the adequate or reasonable interval of time to decide a repeat biopsy: 

“Consider a repeat clinical and US examination and TSH measurement in approximately 12 months in accordance with the clinical setting [BEL 3, GRADE B]. https://doi.org/10.4158/ep161208.gl

Moreover, in case of no US or FNA modifications from the previous nodule assessment:

“In asymptomatic nodules with a repeated benign cytology and no suspicious clinical or US features, routine follow-up may be avoided”, BEL 3, GRADE B; https://doi.org/10.4158/ep161208.gl

If it is true (as reported in limitations) that a minimum follow-up time of 12 months could potentially lead to loose some malignancies (although this evenience is highly improbable), in general we can infer from these recommendations that nodules with a benign cytology and no clinical, radiological or cytological modifications can be reasonably considered BENIGN, thus entering in the evaluation of both US and FNA systems prediction. Moreover, considering the cases with available histology separately would introduce a further bias due to the relative low number of TIR2/BENIGN cases with an over-representation of the malignant ones, with obvious consequences on the statistical robustness of such evaluation.

11. In Figure 4 the headings are not clear and the Reviewer cannot assess neither the x axis nor the y axis.

Thanks for pointing this out. We provided a new picture in which the headings are appropriately readable

12. The Authors state in the Discussion section that ”The sensitivity and specificity of the SIAPEC system were 94.1% and 97.5% respectively, coherent with literature data (93-100% and 54-71%)”. Is this computed from the dataset of the study and applies to the institutional sensitivity and specificity of the SIAPEC? It should then be moved then to the Results section.

Thanks for your prompt observation. The reported 94.1% and 97.5% sensitivity and specificity of cytology have been evaluated on our cohort, as reported in RESULTS just before Table 3 where the performances of the ACR system are reported as well. Moreover, the SIAPEC system doesn’t have a specific institutional sensitivity or specificity, being a classification system mainly based on the “risk of malignancy” (ROM) concept (see comment 8 of the #1 reviewer for a more extensive discussion of the topic). Thus, we reported the comparison of the performances of this system on our cohort with a recent review on the argument (https://doi.org/10.3389/fendo.2020.00044).  

13. Please expand on the study limitations in the discussion section.

Thanks, addressed.

Reviewer 2 Report

the paper is clinically relevant with regard to the role of risk stratification of thyroid nodules for further work, i.e. FNA cytology. 

Comments: the study populations should be detailed with regard to thyroid function, i.e. euthyroidism of (subclinica) hypothyroidism which migh substantially guide the interpretation of nodules, but also their functional assessment by thyroid scintigraphy as recommended in ATA but even european guidelines

2.3. details the criteria for further work-up at follow up visits - how many patients exhibited significant changes, particuarly in size, according to these criteria. Please also detail the course of benign and if detected malignant nodules over time later on in the Results section

49 patients with cancer were detected: it might be of interest to characterise these patients with regard to histopathological TN(M) stage, TIRADS and FNA cytology findings. Notably, the proportion of papillary thyroid cancer seems low - please comment on. 

One might calculate a ROC based  on TIRADS >2  findings to determine the optimal threshold for the size of nodules subjec to FNA.

I suggest to present a graphy on Bland-Altmann analysis to illustrate the range of nodule size covered in that analysis. 

Conclusion section I suggest to comment on the presumed reduction in scores for nodule composition and hypoechogenicity in the Discussion section, only. 

Reference 19 is given in upper-case letters,only

 Reference 24 is incomplete

Author Response

1. the paper is clinically relevant with regard to the role of risk stratification of thyroid nodules for further work, i.e. FNA cytology. 

Thanks for having evaluated our work and for the appreciation you demonstrated on its clinical relevance. We hope that the further improvements prompted by the suggestions made by you and by the other reviewer will further improve the quality of this work.

2. Comments: the study populations should be detailed with regard to thyroid function, i.e. euthyroidism of (subclinica) hypothyroidism which migh substantially guide the interpretation of nodules, but also their functional assessment by thyroid scintigraphy as recommended in ATA but even european guidelines

Thanks for your observations. Actually we added in METHODS the evaluation of thyroid function by the endocrinologist and the eventual nuclear medicine study with scintigraphy where appropriate following the guidelines. Moreover, the detailed distribution of EU/HYPO/HYPER thyroidism in our cohort has been reported in the first section of RESULTS, specifying that none of the nodules studied by means of scintigraphy demonstrated an hyperfunctioning (HOT) state.

3. details the criteria for further work-up at follow up visits - how many patients exhibited significant changes, particuarly in size, according to these criteria. Please also detail the course of benign and if detected malignant nodules over time later on in the Results section

Thanks for your advice. In the analyzed cohort, as already stated in methods, the cases classified as TIR1C and TIR2 performed an US examination 12 months after the first US-guided FNA by the same radiologist. None of these cases demonstrated on re-evaluation the appearance of any of the following features:   

  • new echographic malignant features       
  • >20% increase in size           
  • nodes metastasis          
  • appearance of new suspicious nodules

Thus these cases were considered benign as per previous recommendations In asymptomatic nodules with a repeated (12-months) benign cytology and no suspicious clinical or US features, routine follow-up may be avoided”, BEL 3, GRADE B; https://doi.org/10.4158/ep161208.gl

On the other hand, histological evaluation was carried out on 10.2% of nodules (49/480), the majority of which (81.6%) belonging to the group of cases with a FNA result ≥TIR3B (40/49). The remaining patients with a TIR3A class that did not undergo surgery were followed-up with the US and a minority of these (17/96) showed modification of either nodule shape or dimension, leading to a repeat biopsy with confirmation of TIR3A class (n=10) or downgrading to the TIR2 class (n=7).

.

4. 49 patients with cancer were detected: it might be of interest to characterise these patients with regard to histopathological TN(M) stage, TIRADS and FNA cytology findings. Notably, the proportion of papillary thyroid cancer seems low - please comment on. 

With regard to your comment, we added a characterization of the subset of malignant cases assessed by histology with a supplementary table 1 in the first part of results, which contains the histological subtype of the neoplasm, the TNM stage and the TIRADS FNA classes. Moreover, answering to your second observation, according to the literature only 5-15% of nodules are malignant carcinomas (https://doi.org/10.1016/j.maturitas.2016.11.002) and among them PTCs account for about 65-93% (WHO 2017). In our case series, the malignant carcinomas are 7% (34/480) and - considering only malignant nodules - the prevalence of PTCs is 79% (27/34). We addressed this point changing the percentage in RESULTS from 53% (26/49) to 79% (27/34).

5. One might calculate a ROC based on TIRADS >2  findings to determine the optimal threshold for the size of nodules subjec to FNA.

We appreciate your suggestion, but we think that this issue deserves an ad-hoc study and the findings of the present paper represent a preliminary step to that aim. Due to the relevance of this point, the following sentence has been added in the Conclusion section: “An update of the size criteria and of the US features points might be considered in future studies to increase the screening performances of the ACR-TIRADS system.”

6. I suggest to present a graphy on Bland-Altmann analysis to illustrate the range of nodule size covered in that analysis. 

We have followed the reviewer suggestion and added into the supplementary material the Bland-Altman graph to illustrate the results reported in the paper on the agreement between the measures of the same nodule provided by two radiologists.

7. Conclusion section I suggest to comment on the presumed reduction in scores for nodule composition and hypoechogenicity in the Discussion section, only. 

ADDRESSED

8. Reference 19 is given in upper-case letters,only

The reference has been changed to lower-case letters.

9.  Reference 24 is incomplete

The reference has been corrected.

Round 2

Reviewer 1 Report

Although the Authors have made efforts to respond to the Reviewers comments,  there is generally a lack of progress in the revision of the manuscript. In my opinion, the Authors have not improved their manuscript adequately and do not provide a clinical comprehensive message on the clinical utility of ACR-TIRADS to the readership of Cancers. In addition, there is a linguistic issue that makes the language of the manuscript hard to follow.

Reviewer 2 Report

Thank you for answering the comments and questions raised